# How the Chemical Properties of GBCAs Influence Their Safety Profiles In Vivo

**DOI:** 10.3390/molecules27010058

**Published:** 2021-12-23

**Authors:** Quyen N. Do, Robert E. Lenkinski, Gyula Tircso, Zoltan Kovacs

**Affiliations:** 1Department of Radiology, University of Texas Southwestern Medical Center, 5323 Harry Hines Boulevard, Dallas, TX 75390, USA; Quyen.Do@utsouthwestern.edu (Q.N.D.); Robert.Lenkinski@utsouthwestern.edu (R.E.L.); 2Department of Physical Chemistry Debrecen, University of Debrecen, Egyetem tér 1, H-4032 Debrecen, Hungary; Gyula.Tircso@science.unideb.hu; 3Advanced Imaging Research Center, University of Texas Southwestern Medical Center, 5323 Harry Hines Boulevard, Dallas, TX 75390, USA

**Keywords:** gadolinium-based contrast agents, thermodynamic stability, kinetic inertness, gadolinium deposition, *T*_1_ hyperintensity

## Abstract

The extracellular class of gadolinium-based contrast agents (GBCAs) is an essential tool for clinical diagnosis and disease management. In order to better understand the issues associated with GBCA administration and gadolinium retention and deposition in the human brain, the chemical properties of GBCAs such as relative thermodynamic and kinetic stabilities and their likelihood of forming gadolinium deposits in vivo will be reviewed. The chemical form of gadolinium causing the hyperintensity is an open question. On the basis of estimates of total gadolinium concentration present, it is highly unlikely that the intact chelate is causing the *T*_1_ hyperintensities observed in the human brain. Although it is possible that there is a water-soluble form of gadolinium that has high relaxitvity present, our experience indicates that the insoluble gadolinium-based agents/salts could have high relaxivities on the surface of the solid due to higher water access. This review assesses the safety of GBCAs from a chemical point of view based on their thermodynamic and kinetic properties, discusses how these properties influence in vivo behavior, and highlights some clinical implications regarding the development of future imaging agents.

## 1. Introduction

Contrast agents in diagnostic magnetic resonance imaging (MRI) or magnetic resonance angiography (MRA) are intravenous drugs used to enhance the contrast of MR images for clinical diagnosis and disease monitoring. Gadolinium-based contrast agents (GBCAs), the most widely used MRI contrast agents, have been instrumental in research and clinical applications for the detection of various pathologies including cancer, infections, bleeding, and neurological disorders. Since approval of the first GBCA in 1988, Bayer estimated 450 million doses have been administered to patients worldwide [1].

All clinically approved GBCAs are neutral or negatively charged with nearly identical biodistribution. GBCAs are generally extracellular fluid agents and rapidly equilibrate in the extracellular space. The pharmacokinetic parameters (clearance rate, distribution and elimination half-life and steady state distribution volume) that characterize the distribution and elimination of the agent can be determined by fitting the plasma concentration vs. time data to the standard two-compartment open model. As expected, based on their highly hydrophilic nature, GBCAs are eliminated by renal filtration with a half-life of approximately 90 min in healthy individuals [2,3,4,5,6,7,8,9,10]. It is worth noting that agents with hydrophobic aromatic substituents (benzyloxymethyl or p-ethoxybenzyl) on the ligand framework are also partly cleared by hepatobiliary excretion, and consequently, these complexes are used as liver specific agents (Eovist, Multihance (Europe)).

Although longer than normal, elimination half-life was noticed early on in patients with renal failure; this had not raised any concerns until about a decade later. The first published report describing nephrogenic systemic fibrosis (NSF) appeared in 2000 [11]. NSF is a devastating systemic disease characterized by the formation of scar tissue (fibrosis). Connection between kidney failure and contrast-enhanced MRI in the etiology of NSF was made in 2006 [12,13]. Reports of GBCA-linked NSF led to safety-related labeling changes with class warning and contraindications in patients with poor renal function. These safety measures eliminated NSF altogether. However, the safety of GBCAs came under close scrutiny again in 2014, when high MRI signal intensity on unenhanced *T*_1_-weighted brain images of patients who had repeated exposure to GBCAs was described [14]. This observation was confirmed in numerous subsequent studies [15,16,17,18] and the presence of gadolinium deposition was confirmed using inductive coupled plasma mass spectroscopy (ICP-MS). These reports led to a new class warning for all GBCAs and restriction of the use/suspension of the authorization of some linear GBCAs by the U.S. Food and Drug Administration (FDA) and European Medicines Agency (EMA). In their statements, the FDA and the International Society of Magnetic Resonance in Medicine (ISMRM) [19] stressed other than in NSF, that there is currently no evidence that gadolinium deposition in the brain and other tissues has caused any harm to patients.

It is our belief that despite its shortcoming, the extracellular class of GBCAs is still among the safest drugs ever introduced. However, the preclinical and clinical reports on GBCA deposition in various tissues indicate that there is room for improvement and these observations should be taken into account in the design of future GBCAs. This review assesses the safety of GBCAs from a chemical point of view based on their thermodynamic and kinetic properties, discusses how these properties influence in vivo behavior, and highlights some clinical implications regarding the development of future imaging agents.

## 2. Thermodynamic Stability and Kinetic Inertness of GBCAs

Gd^3+^ uniquely has a symmetric seven electron ground state (^8^S_7/2_) which imparts long electronic spin relaxation time (*T*_1*e*_) to these unpaired electrons, making it highly efficient at enhancing both the longitudinal (*R*_1_) and transverse (*R*_2_) relaxation rates (*R_i_* = 1/*T_i_*, *i* = 1,2, where *T* is the relaxation time) of water proton spins. Gd^3+^ is primarily utilized to shorten the longitudinal (*T*_1_) relaxation time; therefore, GBCAs are known as *T*_1_ contrast agents. Gd^3+^ has no natural biological role. It has an ionic radius of 0.99 Å, approximating the size of Ca^2+^ but with a higher charge. It can thus efficiently compete with Ca^2+^ in biological systems for Ca^2+^-binding enzymes, disrupting critical Ca^2+^-signaling pathways. In pH > 6 solution, free Gd^3+^ either hydrolyzes to insoluble oxides and hydroxides or quickly complexes with ions such as carbonate and phosphate to form stable precipitates. In the design of contrast agents for MRI, Gd^3+^ is enclosed in a ligand to form a complex that is expected to remain chelated in the body and be excreted intact. Nine GBCAs have been approved by FDA. All ligands used as components of approved GBCAs are based on two octadentate polyaminocarboxylate type chelators: DOTA (macrocyclic) and DTPA (linear) (Figure 1). Gd^3+^ has a coordination number of 9; thus, all GBCAs contain a metal-bound inner sphere water molecule occupying the ninth coordination site. Although the relaxivity, a measure of the GBCA’s effectiveness at enhancing the relaxation times of the water protons, is proportional to the number of coordinated water molecules (*q*), increasing *q* can result in a decrease in complex stability. Currently, all approved GBCAs (Figure 2) have only one inner sphere water molecule. Gadopiclenol, a macrocyclic pyclen-based contrast agent going through clinical trial, is an exception with a *q* = 2 (Figure 3). The exact prediction of complex stability in vivo is challenging. Physical properties such as thermodynamic and kinetic stabilities gathered from in vitro and in vivo distribution data allow a certain degree of prediction and explanation for the in vivo behavior of GBCAs.

The thermodynamic stability of the complex is characterized by the thermodynamic stability constant, which is the equilibrium constant for the reaction between the metal ion and the fully deprotonated ligand (Equation (1)).
(1)$$Ln^{3+}+L⇌LnL\;K_{LnL}=\frac{\left[LnL\right]}{\left[Ln\right]\left[L\right]}$$

However, in aqueous solution, depending on the pH and its protonation constants (basicity), the ligand will be fully or partially protonated. This means that the formation of the metal complex is essentially a competition between protons and the metal ion for the basic donor sites of the ligand. It is also important to emphasize that the true thermodynamic stability of a complex at a given pH is characterized by the conditional stability constant (*K^C^*), which takes into account the protonation of the ligand. Thus, taking into account ligand protonation, the thermodynamic stability of a lanthanide chelate with one metal-bound water molecule (inner sphere water molecule) is characterized by Equation (2):(2)$$Ln{\left(H_{2}O\right)}_{8}^{3+}+H_{n}L⇌LnL\left(H_{2}O\right)+nH^{+}+7H_{2}O\;K_{LnL}^{C}=\frac{\left[LnL\right]}{\left[Ln\right]{\left[L\right]}_{total}}=\frac{\left[LnL\right]}{\left[Ln\right]\left[L\right]\propto_{H}}=\frac{K_{LnL}}{\propto_{H}}$$
where [*L*]*_total_* is the total concentration of the free and protonated ligand species which is not bound to the lanthanide ion and α_*H*_ is the total/equilibrium ligand concentration ratio and can be expressed using the pH and the protonation constants of the ligand as *α_H_* = 1 *+ K*_1_[*H*^+^] + *K*_1_*K*_2_[*H*^+^]^2^
*+ K*_1_*K*_2_…*K_n_*[*H*^+^]^*n*^. Thus, if the ligand is protonated at a particular pH, the conditional stability of a complex at that pH will always be lower than its thermodynamic stability (*K^C^* < *K*) and will decrease as the solution becomes more acidic and the ligand undergoes further protonation. Therefore, the thermodynamic stability constant is inadequate by itself to predict the behavior of GBCAs. In the case of Omniscan, the relatively low conditional stability constant prompted addition of excess ligand as the calcium complex of DTPA-bis(methylamide) (Caldiamide), into Omniscan formulation to shift the equilibrium to the complex formation. Despite this, Omniscan is tied to the largest number of NSF cases and has higher gadolinium levels remaining in the body after administration than any other GBCA [20,21]. There is a more or less linear relationship between the basicity of the ligand donor atoms and the thermodynamic stability of the resulting metal complexes as more basic ligands generally form more stable complexes [22]. Although both linear and macrocyclic ligands used in GBCAs have 8 donor atoms, macrocyclic complexes exhibit a slight increase in thermodynamic stability compared with the linear ones. This is known as the macrocyclic effect. Macrocyclic ligands with a 12-membered DOTA chelate ring structure are inherently more pre-organized than their open-chain analogs with an internal cavity with oxygen and nitrogen binding sites. The free macrocyclic ligand has the same or nearly the same solution conformation as the gadolinium-bound chelate. It takes less energy for the macrocyclic ligand to convert into the conformation necessary for complex formation than the linear ligand. The basic mechanism for the complex formation of polyamino polycarboxylate ligands with trivalent metal ions has been recognized for over 50 years [23]. The negatively charged carboxyl groups of the ligand rapidly displace some of the inner sphere water molecules of the metal ion and results in the formation of a protonated, “out-of-basket” intermediary complex in which the metal is coordinated by only the carboxylates of the ligand and water molecules. This is followed by the deprotonation and concomitant rearrangement of the protonated intermediate to the final “in-cage” complex. This latter step determines the rate of the entire process. It is generally very fast for flexible open-chain ligands such as EDTA and DTPA that can easily wrap around the metal ion [24]. However, the deprotonation and simultaneous rearrangement of the intermediate can be rather slow for rigid ligands. These include rigidified open-chain chelators such as *trans*-cyclohexanediamine tetraacetic acid (CDTA) and cyclen (1,4,7,10-tetraazacyclododecane) derivatives such as DOTA. Although the complexation reaction with linear ligands happens in a matter of seconds or milliseconds, macrocyclic complexation is a more time-consuming process and can take several hours or even days depending on the structure of the ligand and reaction conditions (pH, temperature) [22,25,26]. 

The kinetics of dissociation for a lanthanide complex (*LnL*) is described by Equation (3).
(3)$$-\frac{d{\left[Ln\left(L\right)\right]}_{t}}{dt}=k_{obs}{\left[Ln\left(L\right)\right]}_{t}$$

Kinetic inertness is characterized by the experimentally observed rate constant of dissociation, *k_obs_* or the corresponding half-life (*t*_1/2_). For a first order reaction, *t*_1/2_
*=*
*ln*2/*k_obs_*. There are several possible mechanisms for the dissociation. These include spontaneous, acid catalyzed, base catalyzed, metal ion assisted (transmetallation) and ligand assisted pathways [27,28]. Transmetallation describes the displacement of the lanthanide from its chelate by another metal ion. In general, open-chain complexes dissociate much faster than macrocyclic ones. In vivo, linear complexes undergo dissociation by acid catalyzed, endogenous metal ion (Zn^2+^, Cu^2+^, Ca^2+^) and ligand (phosphate, carbonate, citrate) assisted mechanisms. The macrocyclic complexes are much more inert. They dissociate predominantly by proton assisted dissociation even at pH 7. The first step in the proton assisted dissociation is the protonation of the complex, generally at one of the oxygens of a coordinated carboxylate. This is followed by the transfer of the proton to one of the nitrogens to form a protonated intermediate in which the metal is coordinated to the carboxylates only [22]. This intermediate then can either undergo full dissociation or can reform the complex [25]. Dissociation kinetic studies are usually performed in strongly acid solutions often under different experimental conditions. There is also a slight discrepancy in the reported stability constant values, depending on different experimental conditions used by different research groups. Despite the lack of standardization, the universal observation is that the kinetic stability of the FDA approved macrocyclic chelates is far superior to that of the linear chelates. The significantly higher kinetic inertness of macrocyclic GBCAs compared with the linear GBCAs undoubtedly contributes to the lower retention of macrocyclic GBCAs in tissues.

Thermodynamic and kinetic properties of approved GBCA are summarized in Table 1.

## 3. When Does Kinetic Inertness Matter?

Upon entering the body, GBCAs are exposed to endogenous metal ions (Cu^2+^, Zn^2+^), proteins, and biologically available anions such as phosphates and carbonates, all of which have the potential to assist in gadolinium complex dissociation. To minimize the dissociation, gadolinium complexes must be kinetically inert under this condition. In healthy humans, GBCAs have an elimination half-life of ~90 min [2,3,4,5,6,7,8,9,10]. The short plasma residence time means that thermodynamic equilibrium is not established, and kinetic inertness determines whether the complex would dissociate at this time. Models developed to predict in vivo fate of GBCA based on only thermodynamic stability constants especially fail when applied to macrocyclic GBCA [39,40,41,42]. They provide an incomplete picture and also misguided prediction: based on thermodynamic stability, Multihance and Magnevist are predicted to be slightly more stable than Gadavist, a macrocyclic agent, an observation not supported by both in vitro and in vivo experiments [43,44]. The results of these in vivo preclinical experiments were similar to human studies [21,45] where they studied the concentration of GBCAs in bone of patients who underwent hip arthroplasty with prior administration of Prohance or Omniscan. Omniscan had significant higher bone tissue retention than Prohance. Gadolinium deposition reports are in agreement with in vitro dissociation kinetic data. In an experiment measuring gadolinium released in human serum at 37 °C, the amount of gadolinium disassociation was found to follow this order: DTPA bis(aminde)s > DTPA and side arm substituted DTPA > backbone substituted DTPA. Predictably, there was negligible gadolinium release from macrocycic agents.

However, thermodynamic stability plays a more important role in pathological conditions where GBCAs excretion is affected (e.g., patients with impaired kidney function) and the complex stays in the body longer than in healthy individuals [46]. The importance of thermodynamic stability is well illustrated in experiments [47] where Ominiscan, Magnevist, and Dotarem were incubated in cell culture medium (0.91 mM phosphates) for 24 h. The long incubation time allowed a thermodynamic equilibrium to be established between the free Gd^3+^ and the ligand. The amount of free metal released and precipitated as insoluble phosphate salt was exclusively determined by the thermodynamic stability constant. Omniscan with the lowest thermodynamic stability produced a larger amount of precipitation and thereby exhibited a higher degree of dissociation than Magnevist and Dotarem. Therefore, we believe that in situations when a pseudo-equilibrium can be established in vivo, complexes with low thermodynamic stability may generate more gadolinium deposit than ones with higher thermodynamic stability.

## 4. What Structural Features Govern Kinetic Inertness?

Over the years, a substantial amount of research has been invested into studying the factors that influence the kinetic properties of lanthanide complexes. As a result, it is now clear that the kinetic behavior of the complex is determined by the rigidity of the ligand framework and the presence and basicity of possible protonation sites in the complex. A more rigid ligand framework translates into higher kinetic inertness because the conformational changes necessary for the release of the metal ion from the coordination cage take place slower in a more rigid structure [48]. All DTPA and DOTA based complexes contain five membered chelate rings, which are essential to the overall rigidity. Six membered chelate rings can be incorporated into the structure by inserting an extra CH_2_ unit into the ligand backbone or sidearm. This results in faster exchange rate of the inner sphere (metal-bound) water molecule and thereby higher relaxivity because of the increased steric hindrance around the metal-bound water molecule. However, this structural modification generally also results in a significant decrease in the kinetic inertness because 6-membered chelate rings are more flexible than the 5-membered ones. Chelate rings (not to be confused with the macrocyclic ring of DOTA) are structural units composed of the metal ion, two ligand donor atoms, and a bridge linking the donor atoms, usually a 2 or 3-carbon chain for 5- and 6-membered chelate rings, respectively [49,50].

The higher kinetic inertness of DOTA based complexes is largely due to the more rigid macrocyclic backbone of DOTA in comparison with the flexible open DTPA derivatives. The most common way to improve kinetic inertness of both linear and macrocyclic complexes is the introduction of substituents to the ligand backbone and/or acetate sidearms. These substituents increase rigidity by increasing the steric bulk as well as by imparting chirality to the ligand backbone and/or sidearm, which results in a preference for a specific coordination geometry. These effects slow down internal conformational motions thereby increasing kinetic inertness [48]. The effect of substitution is well illustrated by the improved kinetic inertness of Multihance and Eovist over Magnevist (Table 1). Likewise, the rigidity of DOTA can significantly be enhanced by placing alkyl substituents on the macrocyclic and/or sidearm methylene carbons [51,52,53]. The effect of these substitutions on the kinetic inertness of DOTA complexes is much more dramatic in comparison with the open-chain DTPA chelates. For example, placing four ethyl substituents in S-configuration at 2, 5, 8, and 11 positions of the macrocyclic ring in Gd-DOTA prevents the interconversion of the square antiprism (SAP) and twisted square antiprism (TSAP) coordination isomers, thereby allowing their separation and characterization. The kinetic inertness of these complexes was measured in 1 M HCl and compared with that of Gd-DOTA. Virtually no decomplexation was observed for the SAP isomer and only minimal amount of metal was released from the TSAP isomer after 487 h. The half-life of Gd-DOTA under these acidic conditions was about 25 h [54]. A related approach for increasing the rigidity of the complex, albeit without inducing chirality, is the incorporation of a pyridine ring into the ligand backbone. It was shown some time ago that lanthanide complexes of the heptadentate ligand PCTA (3,6,9,15-tetraazabicyclo[9.3.1]pentadeca-1(15),11,13-triene-3,6,9-triacetic acid), which contain a pyridine moiety fused to the cyclen ring, have surprisingly high kinetic inertness for a bishydrated chelate [55]. Both of these principles (pyridine fusion and sidearm substitution) were applied in the design of Gadopiclenol, a high relaxivity GBCA currently in human clinical trials [56,57,58,59] to achieve excellent kinetic stability. A 2- to 3-fold greater relaxivity enhancements compared with current GBCA were achieved through a combination of two coordinating water molecules and slower molecular motion [57].

For DTPA-based GBCAs, a similar improvement in inertness can be achieved by attaching substituents to either the acetate arms or diethylenetriamine backbone of the DTPA ligand. As mentioned earlier, in the cases of Ablavar, Multihance, and Primovist/Eovist, the hydrophobic substituents also alter the biodistribution of the agent and improve the relaxivity via interaction with serum albumin.

The first step in acid catalyzed dissociation is the protonation of an oxygen in the coordinating sidearm followed by proton transfer to the nitrogens. Therefore, unlike thermodynamic stability, the kinetic inertness usually decreases with increasing ligand basicity as this enhances the rate of proton-assisted dissociation. A good example is the gradual decrease of kinetic inertness upon stepwise substitution of methylenephoshonate coordinating sidearm for the acetates in DOTA complexes. The coordinated phosphonate group can be easily protonated on the noncoordinating oxygen in the pH range 5–7, which promotes acid catalyzed dissociation [60,61]. The opposite effect can be observed when in DOTA-tetraamide complexes: the half-life of dissociation dramatically increases when the carboxylates are replaced with carboxamides (acid-catalyzed dissociation rates of lanthanide DOTA-tetraamides are 25- to 30-fold slower than those of the corresponding DOTA complexes) [62,63].

## 5. What Prediction Can We Make about Gadolinium Deposition? How Do Those Predictions Relate to Preclinical Results?

The pharmacokinetics and physiological profiles of GBCA are related to the chemical structures of the GBCA, specifically linear or macrocyclic. The linear GBCAs have flexible chelators whereas the macrocyclic agents have rigid caged structures. The higher flexibility of the linear agents results in more rapid dissociation of gadolinium from the chelate and a higher likelihood of transmetallation than the macrocyclic agents. In the case of NSF, virtually no new cases have been reported over the last decade. This is partly due to recommendations made by the American College of Radiology and other organizations about a reduction in the use of linear GBCAs that have been associated with the most cases of NSF [64,65]. Based on what we know about the chemistry of these complexes, we could predict that macrocyclic GBCAs would likely be cleared as intact chelates rather than deposited in the brain, whereas linear chelates may be more likely to dissociate and potentially leave gadolinium-containing deposits. In situations where GBCAs secretion is limited and a pseudo-equilibrium establishes, the degree of GBCA retention would be strongly influenced by the thermodynamic stability.

It was previously thought that GBCAs do not cross the blood–brain barrier and are cleared from the brain via venous drainage. From preclinical studies, we now know that regardless of the chemical structure, all GBCAs enter the rat brain intact [66,67,68,69,70,71,72,73]. There are several proposed pathways on how GBCAs enter the brain: directly from the blood via the blood–brain barrier or move from the blood to cerebrospinal fluid and then enter the brain parenchyma via the ependyma or the perivascular pial-glial basement membranes system [74]. A number of publications suggest the involvement of the glymphatic system [75,76,77,78,79]. The glymphatic system is responsible for waste removal (soluble proteins and metabolites) as well as distribution of glucose, lipids, amino acids, growth factors, and neuromodulators in the central nervous system by utilizing a network of paravascular channels formed by astroglial cells. The cerebrospinal fluid (CSF) entering these paravascular spaces mixes with interstitial fluid (ISF), and thereby directs waste products into lymphatic vessels and subsequently clears them from the body [80,81,82]. Evidence seems to indicate that although all GBCAs enter intact, macrocyclic GBCAs are subsequently cleared from the brain without appreciable dissociation, whereas linear agents are more likely to dissociate and leave behind residual gadolinium. There is a significant association between the concentrations of GBCA administration, chelate subtype (macrocyclic vs. linear) and the extent of deposition. Macrocyclic agents have diminished elemental gadolinium tissue deposition compared with linear agents. Within linear agents, Robert et al. studied the trend of gadolinium deposition in brain of healthy rat and found the following trend: Omniscan > Magnevist > MultiHance [68]. This trend follows the thermodynamic stabilities of these compounds. Lohrke et al. observed a similar trend when studying rat brain and skin [66]. Within macrocyclic agents, there are more clinical and preclinical reports of MRI and histopathologic findings associated with gadolinium retention for Gadovist than for Dotarem and Prohance [83]. Animal studies performed with repeated administration (up to 20 administrations overall, 4 injections/week over 5 weeks) of high doses (0.6 mmol/kg) of macrocyclic agents (ProHance, Dotarem, Clariscan, and Gadovist) [84,85] revealed that gadolinium levels measured by ICP-MS in the rat brain were significantly lower after cumulative administration of ProHance (gadoteridol) than the rest of the agents [Dotarem/Clariscan (gadoterate meglumine), and Gadovist (gadobutrol)]. It should be noted that these experiments were performed using much higher amounts of GBCAs than the clinical dosages. Therefore, the observations may not represent a clinical scenario. The mechanism by which macrocyclic agents would be retained is unknown, although a recent report on interaction of macrocyclic GBCAs with collagen [86] suggests that binding of intact macrocyclic complexes to components of the extracellular matrix may account for the retention of intact macrocyclic GBCAs complexes. This study showed that all macrocyclic agents have similar affinity to collagen type I, but the maximum amount of collagen bound was slightly different: it decreased following the order of gadoterate meglumine > gadobutrol > gadoteridol. Collagen is positively charged at pH 7.4 and this trend was attributed to differences in whole or fractional charge at pH 7.4 and/or the hydrogen bonding ability of the complexes. Table 2 summarizes the Gd-deposition data with repeated GBCA exposure. Overall, the majority of imaging findings of *T*_1_ hyperintensity in the dentate nuclei of patients with repeated GBCA exposure are with linear agents and not with macrocyclic agents.

## 6. Mechanism of GBCA Retention and the Chemical Form of Deposited Gadolinium

Since the first reports of an increased signal intensity in non-enhanced *T*_1_-weighted MRI of some brain regions in patients with normal renal function who had previously received multiple doses of GBCAs [14,16,17,100,120,130,131], there have been a number of studies looking into the amount and chemical form of the retained gadolinium in human tissues and preclinical animal models. In the majority of clinical cases, gadolinium retention is detected by *T*_1_-weighted imaging and gadolinium levels are not measured by an analytical method. However, in some cases, the amount of deposited gadolinium in various human tissues has been determined by ICP-MS. These data reveal gadolinium levels in the tens of μg/g tissue range. Christensen et al. found a mean gadolinium concentration of 71.4 ± 89.4 μg/g dry tissue (6.3–348.7 μg/g dry tissue) in the skin of patients with NSF [132]. Non-NSF patients usually have lower levels of gadolinium even after repeated exposre to GBCAs. In human brain autopsy studies, McDonald et al. found that all patients exposed to multiple doses of Omniscan had between 0.1 to 58.8 μg gadolinium per gram of brain tissue [17]. Kanda et al. observed a mean of 0.25 ± 0.44 μg/g of brain tissue from patients who had undergone linear GBCA administration (Magnevist and Omniscan) more than twice [101].

In preclinical studies, GBCAs were injected into rodents at significantly higher dosages than clinical use to study their gadolinium retention/deposition in brain and other tissues. Yet, the deposited gadolinium levels in murine brain tissue were around 1 μg/g wet tissue, somewhat lower than those found in human tissues [69,133]. This may not be surprising as rodents’ renal function/clearance may not be at all similar to those of humans. In addition, the physiological mechanisms of GBCA clearance from the brain are not well understood in either species. Additionally, the comparison between structured studies in rats versus anecdotal reports in humans creates uncertainty in the interpretation. In most animal studies, retained gadolinium concentrations were quantified with inductive coupled plasma mass spectroscopy (ICP-MS) [66,67,68,69,73], transmission electron microscopy (TEM) [73] and laser ablation ICP-MS [134]. Overall, all GBCAs were tested with the exception of Optimark and Ablavar, most likely due to their discontinued status. Across these studies, one can draw the conclusion that the amount of retained gadolinium was much higher in animals exposed to linear agents compared with those exposed to macrocyclic ones. This is in good agreement with the results of human studies. Signal intensity enhancement in the brain on unenhanced *T*_1_-weighted MR images in adults have largely been observed after repeated exposure to linear GBCAs. Similar findings were not reported after serial injections of macrocyclic GBCAs [14,15,16,18,107,120,122,131,135,136]. There is an ongoing debate about a report of enhanced signal intensity after exposure to Dotarem in pediatric population [115,137,138,139,140]. Although this report [115] suggests that GBCA retention in the brain may also occur with macrocyclic agent, it should be kept in mind that this particular study was carried out on pediatric patients who received radiation therapy to the brain. Therefore, blood–brain barrier disruption due to radiotherapy might be involved in the retention of Dotarem [139]. Others observe no enhancement after exposure to macrocyclic GBCAs in the pediatric brain [108,111,119,139,140,141].

Thus, a general picture of gadolinium retention/deposition can be given as follows. As we discussed above, it was hypothesized that all GBCAs enter the rat brain through the glymphatic system intact. Macrocyclic agents are then cleared from the brain over time as intact chelates [15,18,97,118,121,142]. Jost et al. showed evidence of macrocyclic GBCAs continuously excreted from the brain rather than being deposited, similar to linear GBCAs, during a period of 1 year [134]. On the other hand, linear agents are more likely to undergo dissociation and cause gadolinium deposition. The deposited linear agents could be found in various forms: insoluble inorganic salts, intact chelate, and a soluble Gd-macromolecular fraction. Aime et al. and Frenzel et al. suggested the presence of a 300 kDA gadolinium containing macromolecular complex based on gel permeation chromatography (GPC) and ultra-performance liquid chromatography—electrospray mass spectrometry (UPLC-ESI-MS) speciation studies [69,133]. They attributed the observed hyperintensity detected in the *T*_1_-weighted images to the soluble gadolinium containing macromolecular species, arguing that the intact chelate and insoluble inorganic salts would not be able to account for the observed signal enhancement. Although this conclusion is likely to be true, our group showed that insoluble gadolinium compounds could exhibit high longitudinal relaxivities on the surface of the solid due to unrestricted access to water (Figure 4). One can only speculate about the chemical forms of the insoluble and of the highly relaxing gadolinium containing species based on the currently available data in the literature. It is also important to note that the various studies used different experimental protocols regarding type of contrast agents, dosage/injection, total dosage, injection frequency, and sample handling/analysis. In order to compare and build on observations from different groups, it may be important to standardize experimental approaches used in animal studies regarding dosing, injection and tissue harvesting timing, tissue processing, and chemical analytical analysis.

From the point of view of inorganic chemistry, also supported by the experience of NSF, it is plausible that the insoluble species may be gadolinium phosphate. GdPO_4_ is extremely insoluble in water and has a solubility product comparable to the dissociation constant of Gd-DOTA, which has the highest thermodynamic stability among the FDA approved GBCAs [143,144,145]. Analysis of lesional skin and other organs in NSF patients revealed gadolinium deposition [87,88,146]. George et al. examined autopsy skin tissues from a NSF patient using synchrotron X-ray flurorescence (SXRF) microscopy and showed that the insoluble deposits contained gadolinium coordinated in a sodium calcium phosphate material [147]. Xia et al. examined human brain tumor biopsies following contrast-enhanced MR scans in patients without severe renal disease with scanning electron microscopy/energy dispersive X-ray spectroscopy (SEM/EDS) and found insoluble deposits containing gadolinium associated with phosphorus and calcium [148]. However, the situation is complicated by the fact that the morphology of lanthanide phosphates and calcium phosphate doped with lanthanide ions depends on the reagents and experimental conditions used for the precipitation, although some of these clearly not relevant for the in vivo formation of GdPO_4_. The various forms include crystalline solids, gel-like precipitation or colloid nanoparticles and it is quite conceivable that these will have different *T*_1_ or *T*_2_ shortening properties [149,150,151,152,153]. In our hands, a gel-like precipitate of GdPO_4_ showed strong *T*_1_ enhancement on the surface demonstrating that *T*_1_ hyperintensity lesions could contain some form of deposited gadolinium phosphate.

## 7. Clinical Effects of Gadolinum Deposition (-NSF)

At the present time, there are no rigorous studies that have shown association of gadolinium deposition with clinical symptoms or data that suggest that it is harmful to patients. Self-reported clinical symptoms of “gadolinium deposition disease” such as generalized sensory symptoms lack clinical evidence to exclude alternative causes for these symptoms. Published studies [154,155,156] suffered from considerable selection bias and a definite discordance between radiological evidence and individual clinical symptoms.

Gadolinium is considered a pregnancy category C drug, in 2018, the American College of Radiology (ACR) recommends that in pregnant women, “GBCAs should only be used if their usage is considered critical and the potential benefits justify the potential unknown risk to the fetus” [157]. Although there are no well-controlled studies in humans on the effects of GBCA to the fetus, De Santis et al. [158] reported no adverse effect on pregnancy and neonatal outcome in 26 pregnant women who was exposed to GBCAs in the first trimester. In a nonhuman primate model, Prola-Netto et al. [159] showed that even though GBCA, specifically Gadoteridol (Prohance), crosses the placenta, there is a rapid and almost complete clearance of GBCA from fetal circulation back to the mother. The amount of quantified gadolinium in juvenile macaque tissues up to 7 months post-delivery were mostly undetectable.

There are no rigorous clinical studies suggesting a long-term health consequence due to gadolinium deposition/retention. GBCAs were developed to answer a clinical need: to enable a timely and accurate disease diagnosis for better patient care. For a patient with a medical condition requiring evaluation with contrast-enhanced MRI, such as the tens of millions of patients who have received gadolinium contrast safely for years, the risk/benefit ratio currently strongly favors contrast administration.

## 8. Conclusions

In summary, this paper provides a chemical perspective on GBCAs safety assessment. Although there are continuous efforts to design alternative non-gadolinium-based MRI contrast agents (paramagnetic Mn^2+^ or Fe^3+^ complexes and diamagnetic chemical exchange saturation transfer agents) as well as imaging methods that allow clinical diagnosis without contrast administration, it is highly unlikely that gadolinium contrast agents will be replaced in the near future, especially for oncology application. The recent favorable results from Gadopiclenol phase III clinical trials demonstrate value in future GBCA research development. The importance of developing compounds that are kinetically and thermodynamically stable for in vivo application is evident from studies of NSF and gadolinium tissue deposition. Efforts have also been made to lower the administered dose by improving the agents’ relaxivity. The question of what the chemical nature of gadolinium deposits is in regions exhibiting *T*_1_ hyperintensities remains open. However, available data suggest that in case of the kinetically less inert agents, it is likely a form of gadolinium phosphate. A better understanding of gadolinium retention/deposition requires more rigorous and standardized analytical approach in both animal and human studies. This knowledge in turn could provide important directions for the development of safer and more efficient MR imaging agents. It is important to emphasize that at the present time, there are no rigorous studies that have shown any clinical effects of gadolinium retention/deposition in the brain [160].

## Figures and Tables

**Figure 1 molecules-27-00058-f001:**
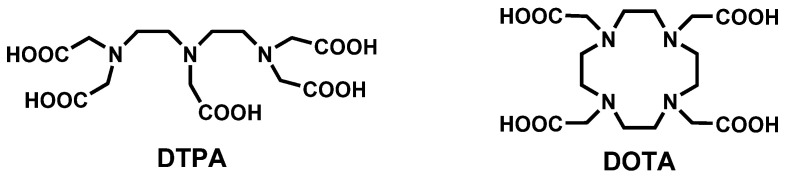
The two standard ligand scaffolds used in clinically approved GBCAs. (DTPA = diethylenetriaminepentaacetic acid or pentetic acid; DOTA = 1,4,7,10-tetraazacyclododecane-1,4,7,10-tetraacetic acid or tetraxetan).

**Figure 2 molecules-27-00058-f002:**
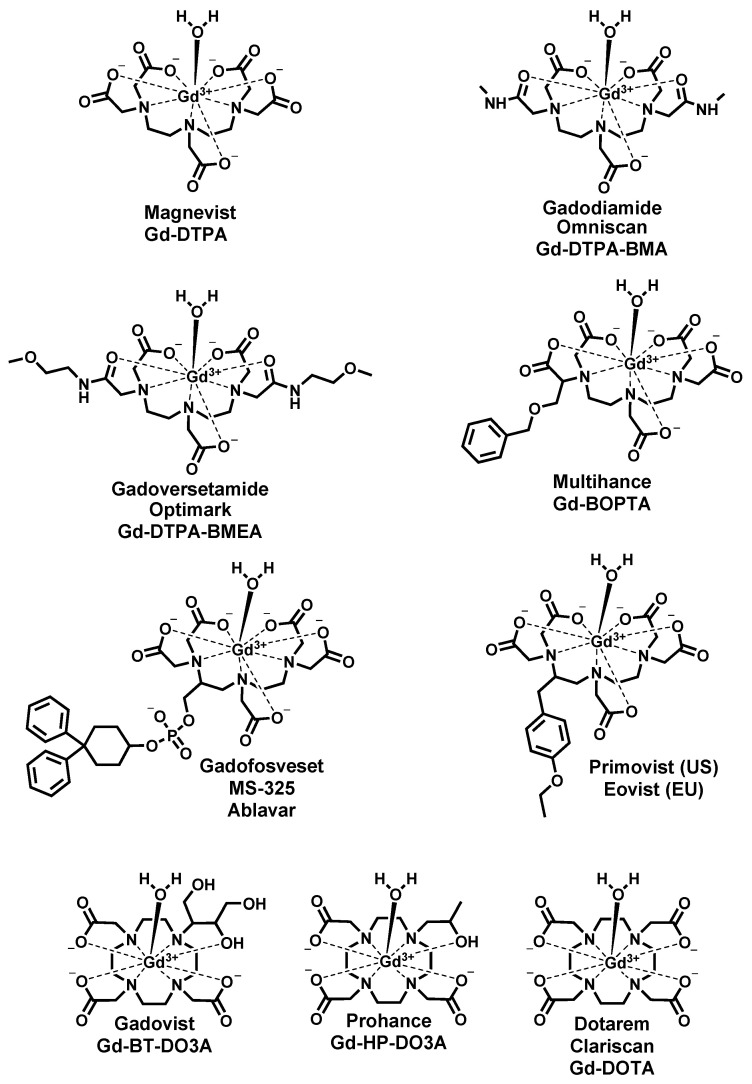
FDA-approved GBCAs utilized in the clinical practice.

**Figure 3 molecules-27-00058-f003:**
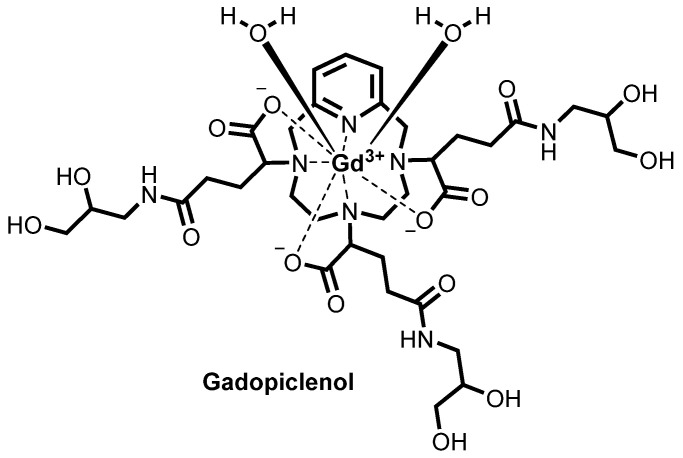
Structure of Gadopiclenol, a macrocyclic pyclen-based contrast agent.

**Figure 4 molecules-27-00058-f004:**
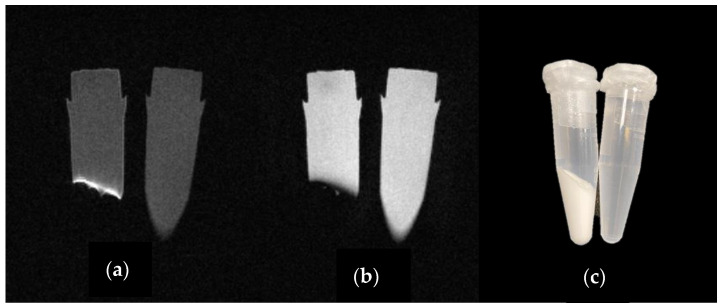
At 1 Tesla, GdPO_4_ phantom shows enhancement at the solution solid interface. (**a**) *T*_1_-weighted spin echo image (TR/TE = 350/13.1 ms), (**b**) *T*_2_-weighted spin echo image (TR/TE = 1600/120 ms), (**c**) A photo of the microcentrifuge tubes making up GdPO_4_ phantom: the tube on the right contains deionized (DI) water, the one on the left contains a GdPO_4_ precipitate in DI water after centrifugation.

**Table 1 molecules-27-00058-t001:** Properties of approved GBCA. The *t*_1/2_ values were calculated for the specific acidic condition of [HCl] = 0.1 M HCl: linear agent condition also has [Zn^2+^] and [Cu^2+^] to account for metal exchange reaction.

GBCA Commercial Names	Common Name	US Approval Year	US Application and Status	Relaxivity at 20 MHz, 25 °C (mM^−1^s^−1^)	Thermodynamic Stability *logK_Gd(L)_* (25 °C)	Kinetic Inertness; *t*_1/2_ (25 °C, 0.1 M HCl)	Formulation ^a^ [29,30]
Magnevist	Gd-DTPA, Gadopentetate	1988	Discontinued; CNS, body, head and neck	4.69 [31]	22.46 (0.1 M KCl) [22]	7.9 × 10^−3^ s [32]	0.5 M, 0.2 mol% excess ligand
Omniscan	Gd-DTPA-BMA, Gadodiamide	1993	CNS, body	4.39 [31]	16.85 (0.1 M NaCl) 16.64 (0.15 M NaCl) [22]	0.66 s [32]	0.5 M, 5 mol% excess ligand
Optimark	Gd-DTPA-BMEA, Gadoversetamide	1999	Discontinued; CNS, liver	5.7 (plasma, 37 °C) [33]	16.84 (0.1 M NaClO_4_) [22]	<5 s [34]	0.5 M, 10 mol% excess ligand
Multihance	Gd-BOPTA, Gadobenate	2004	CNS, MRA	9.7 (plasma, 39 °C) [7] 5.2 [31]	22.59 (0.1 M KCl) 21.91 (0.15 M NaCl) [22]	<5 s [34]	0.5 M, no excess ligand
Eovist Primovist	EOB-DTPA, Gadoxetate	2008	Liver	8.7 (39 °C, 0.47 T) [8]	23.46 [8]	<5 s [34]	0.25 M, 0.5 mol% excess ligand
Ablavar Vasovist	MS-325, Gadofosveset	2008	Product not available; MRA	6.6 (37 °C) [31] 33.4–45.7 (plasma, 0.47 T) [4]	22.06 23.2 (0.1 M Me_4_NCl) [35]	N/A ^b^	0.25 M, 0.13 mol% excess ligand
Prohance	Gd-HP-DO3A, Gadoteridol	2003	CNS, head and neck	3.7 (40 °C) [31]	23.8 [36]	36 years (pH 5.3) [37]	0.5 M, 0.1 mol% excess ligand
Gadovist Gadavist	Gd-BT-DO3A, Gadobutrol	2011	CNS, head and neck	5.2 (plasma 37 °C) [3]	21.8 [38]	65 years (pH 5.3) [37]	1 M, 0.1 mol% excess ligand
Dotarem Clariscan	Gd-DOTA, Gadoterate	2013	CNS (adult, pediatric, neonates)	4.74 [31]	25.6 [9] 24.7 [32]	85 days (pH 2) [32] 37 years (pH 5.3, 37 °C) [37]	0.5 M, 0.1 mol% excess ligand [9] ^c^

^a^ Formulation: Formulation of commercial GBCAs, % excess free ligand. ^b^ *k_obs_* was not reported. The kinetic inertness of Ablavar was estimated to be 10–100 times higher than that of Magnevist from metal exchange reactions [35]. ^c^ Some reports indicate the absence of free ligand.

**Table 2 molecules-27-00058-t002:** Overview of gadolinium depositions from GBCAs in humans.

Agents	Omniscan	Magnevist	MultiHance	Eovist	Ablavar	Optimark	Dotarem	Prohance	Gadovist	Gadopiclenol
NSF risk ^a^	High	High	Intermediate	Intermediate	Intermediate	High	Low	Low	Low	Low ^b^
Human tissue depositions										
Skin	Yes [87,88,89,90]/No [91]	Yes [92,93]/No [91]	Yes [94]					Yes [93]		
Bone	Yes [21,45,95]	Yes [96]	Yes [97]	Yes [97]				Yes [21,45,95,97]	Yes [97,98]	
Brain	Yes [17,99,100,101]	Yes [101,102]	Yes [97,103]	Yes [97]				Yes [97]	Yes [97]	
*T*_1_ hyperintensity in brain tissue	Yes [14,15,16,18,104,105,106]	Yes [14,102,107,108,109,110,111,112]	Yes [18,105]/No [113]	No [114]			Yes [115]/No [16,116,117,118,119]	No [106,117,120]	Yes [121]/No [104,107,112,118,122]	No ^b^

^a^ Based on the EMA recommendation [123,124]. ^b^ Low risk by design [57,58,125,126,127,128], no *T*_1_ hyperintensity in animal models [56,126,129].

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
