# Peer review of "How the Chemical Properties of GBCAs Influence Their Safety Profiles In Vivo"

_molecules, 2021, doi:10.3390/molecules27010058_

Round 1

Reviewer 1 Report

This is the review focused on an important point that has generated a lot of controversy in the past years. The attempt of authors to rationalize this active discussion by providing chemical background on this topic should be highly appreciated. This manuscript definitely deserves publication and should be well received by the scientific community, especially experts in the field.

There are some points that could be considered when revising the current version.

  1. Could authors comment or describe the typical components in the commercial products that are added to the formulations to stabilize the major component, GBCA? For instance, caldiamide is mentioned (l. 118), it might be good to provide its structure.
  2. Is there any influence of osmolarity on the discussed topic (safety of GBCA)? If yes, then the osmolarity should be described (how it depends, how it contributes, etc), otherwise I see no point in listing this property in the Table 1.
  3. Can authors comment on the chirality? E.g. will one or the other stereoisomer ensure higher kinetic stability, will it be equal to the racemic mixture, too? In other words, how important would be to work with (enantiomerically) pure ligands.
  4. Section 4, l. 223-226: please describe what kind of chelate rings are discussed here. Namely, these could be confused with sizes of macrocyclic rings of the chelates (DOTA, DO3A etc).
  5. Section 5, discussion on Gd3+ deposition, ~l. 305-315: would it be possible to provide some quantitative details (amounts of deposited Gd3+), otherwise the current discussion is quite general (too ‘theoretical’), incl Table 2. In addition to Yes/No statements, it would be good to see some numbers in order to get better picture of the discussed process.

The manuscript is generally written well, although there are some places where language could be improved. For instance, section 3 could be read a bit better, right now some of the explanations are a bit confusing.

Some specific points on the language/typos:

  • 110: the sentence and whole new paragraph starts with ‘Where [L]total is…’, which is just continuation of the previous sentence
  • To avoid confusion, I would recommend using Gd3+ when discussing explicitly the ion of gadolinium and then ‘gadolinium complex’ or ‘Gd3+ complex’ instead of ‘Gd complex’. ‘Gd’ is correctly the abbreviation for this chemical element, but usually suggests its neutral state as element/atom. In this context (Gd complex), the abbreviation is used for the noun ‘gadolinium’, not for it as chemical element. To avoid confusion, I recommend changing this expression throughout the manuscript.
  • 197: significant or significantly?
  • 205-207: the sentence starting with ‘In an in vitro experiment…’ does not read well, please rephrase.
  • Figure 4 caption: maybe ‘Eppendorf tube’ can be replaced with another term (e.g. test tube vials), since the Eppendorf is just one specific brand.

Author Response

We thank the Reviewer for the valuable suggestions.  Here is our point-by-point response to the Reviewer's comments:

This is the review focused on an important point that has generated a lot of controversy in the past years. The attempt of authors to rationalize this active discussion by providing chemical background on this topic should be highly appreciated. This manuscript definitely deserves publication and should be well received by the scientific community, especially experts in the field.

There are some points that could be considered when revising the current version.

  1. Could authors comment or describe the typical components in the commercial products that are added to the formulations to stabilize the major component, GBCA? For instance, caldiamide is mentioned (l. 118), it might be good to provide its structure.

The formulation of each commercial contrast agent has been added to Table 1. The structure of the added excess ligand is shown in Figure 1 as part of the gadolinium complex.

  1. Is there any influence of osmolarity on the discussed topic (safety of GBCA)? If yes, then the osmolarity should be described (how it depends, how it contributes, etc), otherwise I see no point in listing this property in the Table 1.

We agree with the Reviewer. Osmolarity does not influence the in vivo stability or gadolinium retention.  Therefore, we removed these data from Table 1 as suggested by the Reviewer.

  1. Can authors comment on the chirality? E.g. will one or the other stereoisomer ensure higher kinetic stability, will it be equal to the racemic mixture, too? In other words, how important would be to work with (enantiomerically) pure ligands.

Chirality was already briefly discussed in the manuscript (page 8, line 386). As we pointed it out, ligand chirality can have a dramatic effect in the stability and kinetic inertness of the corresponding complex by forcing the complex to adopt a specific coordination geometry.  To illustrate this point, we expanded this section by including a specific example (page 8, line 391-400). 

  1. Section 4, l. 223-226: please describe what kind of chelate rings are discussed here. Namely, these could be confused with sizes of macrocyclic rings of the chelates (DOTA, DO3A etc).

To avoid confusion, a definition of chelate rings was included with references (page 8, line 377-380).

  1. Section 5, discussion on Gd3+ deposition, ~l. 305-315: would it be possible to provide some quantitative details (amounts of deposited Gd3+), otherwise the current discussion is quite general (too ‘theoretical’), incl Table 2. In addition to Yes/No statements, it would be good to see some numbers in order to get better picture of the discussed process.

The problem here is that gadolinium retention is usually established by T1 weighted MR imaging and quantitative analysis of gadolinium content is rarely performed. Therefore, we cannot include gadolinium deposition data in Table 2.  However, we included a section discussing reported gadolinium levels measured in biopsy or autopsy samples (page 11, line 520-527).

The manuscript is generally written well, although there are some places where language could be improved. For instance, section 3 could be read a bit better, right now some of the explanations are a bit confusing.

Some specific points on the language/typos:

110: the sentence and whole new paragraph starts with ‘Where [L]total is…’, which is just continuation of the previous sentence

We have corrected the sentence.

To avoid confusion, I would recommend using Gd3+ when discussing explicitly the ion of gadolinium and then ‘gadolinium complex’ or ‘Gd3+ complex’ instead of ‘Gd complex’. ‘Gd’ is correctly the abbreviation for this chemical element, but usually suggests its neutral state as element/atom. In this context (Gd complex), the abbreviation is used for the noun ‘gadolinium’, not for it as chemical element. To avoid confusion, I recommend changing this expression throughout the manuscript.

We have corrected the language as suggested by the Reviewer.

197: significant or significantly?

In the sentence “The significantly higher kinetic inertness of macrocyclic GBCAs as compared to the linear GBCAs undoubtedly contributes to the lower retention of macrocyclic GBCAs in tissues”, we think the correct form is “significantly”.

205-207: the sentence starting with ‘In an in vitro experiment…’ does not read well, please rephrase.

Figure 4 caption: maybe ‘Eppendorf tube’ can be replaced with another term (e.g. test tube vials), since the Eppendorf is just one specific brand.

We have corrected the language as suggested by the Reviewer.

Reviewer 2 Report

In this review, the chemical properties of gadolinium-based contrast agents (GBCAs) based on their thermodynamic and kinetic properties are presented and discussed for possible clinical diagnoses and disease management and highlights some clinical implications related to the development of future imaging agents.

Comments

The subject of the review is interesting and perhaps important for future new applications and/or innovations in the use of contrast agents. The reviewer suggests accepting the manuscript in the present form.

Author Response

Reviewer 2 did not request any changes. We thank the Reviewer for considering our manuscript.